# Serological Vulnerability and Active Infection Detection Among Recently Arrived Migrants in Spain: Results from a Targeted Screening Program

**DOI:** 10.3390/tropicalmed10060169

**Published:** 2025-06-16

**Authors:** Guillermo Lens-Perol, Olalla Vázquez-Cancela, Magdalena Santana-Armas, Angeles Bouzas-Rodriguez, Victoria Tuñez-Bastida, Adrián Domínguez-Lago, Hugo Pérez-Freixo, Cristina Peiteado-Romay, Juan Manuel Vázquez-Lago, Cristina Fernández-Pérez

**Affiliations:** 1Department of Preventive Medicine and Public Health, University Hospital of Santiago de Compostela, Rua da Choupana s/n, 15705 Santiago de Compostela, Spain; 2Health Research Institute of Santiago de Compostela (IDIS), 15706 Santiago de Compostela, Spain; 3Department of Microbiology and Parasitology, University Hospital of Santiago de Compostela, Rua da Choupana s/n, 15705 Santiago de Compostela, Spain

**Keywords:** migrant health, seroprevalence, vaccine-preventable diseases, infectious disease screening, accelerated immunization, public health intervention

## Abstract

Background: Newly arrived migrants are at increased risk for vaccine-preventable and communicable diseases due to low immunization coverage, poor access to healthcare, and challenging migration trajectories. This study describes the implementation and outcomes of a one-stop public health intervention focused on serological screening and accelerated vaccination in recently arrived migrants in Galicia, Spain. Methods: We conducted a cross-sectional descriptive study in July and August 2024 involving 335 adult migrants from sub-Saharan Africa with irregular administrative status and asylum applications. A centralized mobile health unit provided point-of-care screening for immunity against measles, mumps, rubella, varicella, and hepatitis A, alongside testing for active infections, including hepatitis B and syphilis. Sociodemographic and clinical data were collected, and individuals were offered vaccination according to an accelerated immunization schedule. Results: Of 336 migrant adults invited to participate in the study, only 1 individual declined to participate (participation rate: 99.7%). Therefore, 335 migrants were assessed. A significant proportion of participants were susceptible to at least one vaccine-preventable disease, particularly hepatitis B (36.4%, 95% CI 31.3–41.6), measles (22.7%, 95% CI 18.2–27.2), and varicella (16.4%, 95% CI 12.5–20.4). Active infections were detected in 12.9% (95% CI 9.3–16.4) of individuals, including hepatitis B (9.9%, 95% CI 6.7–13.0) and syphilis (3.0%, 95% CI 1.2–4.8). The intervention allowed for timely vaccination and linkage to care, minimizing dependence on passive healthcare access. Conclusions: This study highlights substantial immunization gaps and the presence of undiagnosed infections in vulnerable migrant populations. Centralized and culturally adapted screening programs, combined with accelerated vaccination strategies, are feasible and effective. These findings support the integration of structured protocols into national health systems to ensure equity, reduce transmission risk, and align with WHO and ECDC public health frameworks.

## 1. Introduction

International migration is a growing global phenomenon, driven by complex social, economic, political, and environmental factors. The United Nations defines a migrant as any person who changes their habitual place of residence, regardless of the reason or legal status [1]. In recent decades, Europe has experienced a steady increase in the arrival of migrants, which has posed significant challenges for national health systems. As of 1 January 2024, Spain had registered over 8.8 million foreign-born residents, making it one of the most demographically diverse countries in the European Union [2]. This demographic shift requires health systems to adapt in order to provide timely care, prevention, and epidemiological surveillance, especially for vulnerable subgroups such as refugees, asylum seekers, and individuals with irregular administrative status [3]. Several studies have shown that newly arrived migrants face an increased risk of infectious diseases, many of which are vaccine-preventable [4]. This vulnerability is amplified by incomplete immunization coverage in countries of origin, precarious living conditions during migration, and limited access to healthcare upon arrival in host countries [5,6].

A recent systematic review has highlighted that vaccination coverage among migrants often falls below herd immunity thresholds (HITs), with particularly low rates reported for diseases such as diphtheria (57.4% vs. HIT 83–86%), measles (83.7% vs. HIT 93–95%), and mumps (67.1% vs. HIT 88–93%) [7]. Although these gaps may not represent an immediate threat to local populations, they pose a risk for the reemergence of vaccine-preventable diseases, especially in regions with suboptimal vaccination rates or limited early detection capacity [6,8].

Moreover, there is a notable lack of consolidated data on the immunological status of migrant populations, which hinders effective planning of prevention and control strategies. This limitation is particularly relevant for infections that disproportionately affect migrants, such as tuberculosis, HIV, and hepatitis B [5,6,9]. Despite increasing recognition of these challenges, critical knowledge gaps persist regarding the immune status of newly arrived individuals, their access to vaccination programs, and the structural barriers that limit their inclusion in health systems [10]. These challenges are often more pronounced in non-metropolitan or peripheral regions, where experience in addressing migrant health needs may be more limited.

In this context, the objective of the present study was to assess the serological status and presence of active infections among a group of newly arrived undocumented migrants with pending asylum applications, relocated to Santiago de Compostela, Spain, in July and August 2024. The intervention aimed to identify immunization gaps and inform the development of effective, equity-based public health responses.

## 2. Materials and Methods

### 2.1. Study Design

A descriptive cross-sectional study was conducted using a seroprevalence survey among newly arrived migrants in Santiago de Compostela in July and August 2024. This cross-sectional study was conducted and reported in accordance with the STROBE (Strengthening the Reporting of Observational Studies in Epidemiology) guidelines for observational research.

### 2.2. Selection, Sample, and Procedures

The study population included all migrants from the African continent aged ≥18 years who had recently arrived in Santiago de Compostela, Galicia (northwest region of Spain), and received care at the Monte do Gozo healthcare point during July and August 2024. Recently arrived refers to migrants recently relocated to the study setting, within a context of overall short stay in Spain.

Although the study followed a census approach, including all eligible participants present during the intervention period, a post hoc sample size estimation was conducted based on the 32.1% prevalence of hepatitis B susceptibility reported by Norman et al. [11]. According to standard sample size formulas for prevalence studies (95% confidence level, ±5% margin of error), a minimum of 335 participants would be required. Our final sample met this threshold exactly, ensuring sufficient statistical power, estimated at 80–82%, to generate reliable seroprevalence estimates.

To minimize language and cultural characteristics, all stages of the intervention were supported by trained cultural mediators and interpreters, particularly fluent in French, Arabic (Mauritania), and English. Communication was adapted to the language and comprehension level of each participant to ensure informed consent, accurate data collection, and effective health counseling.

Clinical assessment of the migrant population included a full medical history (anamnesis and assessment of current and previous symptoms and health status) and physical examination. During the initial interview, relevant medical history was collected, and any symptoms present at the time of the clinical visit were documented. The physical examination focused on identifying signs associated with common conditions in this population, such as skin infections or respiratory symptoms.

Demographic, clinical, and serological data were collected from all participants through structured medical assessments and laboratory tests. Demographic data included information on country of origin, age, and migration history. Clinical data encompassed symptom evaluation, physical examination, and documentation of health issues. Serological testing included detection of IgG antibodies for varicella, rubella, measles, mumps, and hepatitis A, as well as hepatitis B markers (HBsAg, anti-HBc, anti-HBs). Additionally, screening for HIV, syphilis, and hepatitis C was performed. Active infection defined by antigen or antibody positivity (e.g., HBsAg or TPAb); past infection by IgG/seroconversion patterns. All tests were performed using validated point-of-care rapid tests and ELISA/CLIA kits approved for clinical use in Spain. Serologies for measles, mumps, and varicella were conducted using Liaison^®^ IgG CLIA assay (DiaSorin S.p.A., Saluggia, Italy). Rubella and hepatitis A/B serology were performed using the ADVIA Centaur^®^ IgG immunoassay (Siemens Healthineers, Erlangen, Bavaria, Germany). Syphilis screening was carried out using the Immulite 2000 Syphilis Screen assay (Siemens Healthineers, Erlangen, Bavaria, Germany). Hepatitis B surface antigen (HBsAg) was detected with the ARCHITECT HBsAg Qualitative II assay (Abbott Diagnostics, Abbott Park, IL, USA). These tests comply with WHO-recommended performance standards. The analytical thresholds (cut-off titers) followed the manufacturer’s instructions and national guidelines.

The selection of diseases included in the screening protocol—measles, mumps, rubella, varicella, hepatitis A and B, syphilis, HIV, and HCV—was based on recommendations from the World Health Organization (WHO) and the European Centre for Disease Prevention and Control (ECDC) for migrant health screening [3,6,8]. These conditions were prioritized due to their epidemiological relevance, vaccine-preventability, and implications for early detection and public health surveillance. As part of the public health intervention, an accelerated vaccination schedule was implemented immediately after serological screening, following the official protocol of the Galician Health Service (SERGAS) for populations with uncertain vaccination status [12]. Vaccines were administered directly based on individual serological profiles, in line with WHO and ECDC guidelines for newly arrived migrants with uncertain vaccination history [3,6,8]. The vaccines included MMR, hepatitis B, varicella, and meningococcal ACWY, among others. MMR and varicella vaccines were given in a two-dose schedule with at least a 28-day interval. Hepatitis B vaccine was administered in a 0–1–6-month schedule, with initial doses provided during the intervention and the remainder scheduled at the referral primary care centers. Meningococcal ACWY and polio (IPV) were administered as single doses according to age criteria. Only the first dose was administered during the intervention; follow-up doses were coordinated through referral to local healthcare centers, with personal vaccination records provided to each participant.

### 2.3. Statistical Analysis

Quantitative variables were summarized using measures of central tendency (mean, median) and dispersion (standard deviation); the mean and standard deviation were used only when the data followed a normal distribution. For categorical variables, absolute and relative frequencies were calculated. Normality was assessed using the Kolmogorov–Smirnov test. Immunological susceptibility rates and prevalence of active infections were expressed as percentages of the total number of participants evaluated. All statistical analyses were performed using IBM SPSS Statistics version 29.0.

### 2.4. Ethical Considerations

This study was conducted in accordance with the Declaration of Helsinki and was approved by the Santiago-Lugo Research Ethics Committee (protocol code 2024/469). As this was a retrospective observational study based on routine clinical activity, the approval was granted in accordance with Spanish Law 3/2018, Additional Provision 17, which regulates the ethical use of health data for research purposes in public health interventions.

## 3. Results

A total of 336 migrants were selected, all of whom were male (100%). Only one individual declined to participate in the study. Therefore, 335 (99.7% participation rate) migrants were assessed. The mean age of the sample was 25.10 ± 6.3 years (range: 18–54 years). All migrants stayed in Spain for 4 months, and in Santiago de Compostela for 1 week. This migrant population originated from nine different African countries. Most participants were from Mali (n = 182; 54.2%) and Senegal (n = 106; 31.5%), followed by individuals from Mauritania, Gambia, Guinea, Côte d’Ivoire, and others with lower representation (Table 1).

During the clinical examination, a total of 89 health complaints were documented (see Table 2).

In the serological analysis, most participants showed immunity to vaccine-preventable diseases. For varicella, 81.8%, 95% CI 77.7–85.9 (n = 274) exhibited detectable IgG levels, while 16.4%, 95% CI 12.5–20.4 (n = 55) lacked immunity. Regarding rubella, 95.8%, 95% CI 93.7–98.0 (n = 321) demonstrated protective antibodies, and 3.0%, 95% CI 1.2–4.8 (n = 10) were identified as seronegative. For measles, 77.0%, 95% CI 72.5–81.5 (n = 258) of participants had evidence of immunity, whereas 22.7%, 95% CI 18.2–27.2 (n = 76) showed no protection. In the case of mumps, 89.3%, 95% CI 85.9–92.6 (n = 299) had protective antibodies, while 9.0%, 95% CI 5.9–12.0 (n = 30) lacked immunity. Additionally, 99.1%, 95% CI 98.1–100 (n = 332) tested positive for anti-hepatitis A (HAV) IgG antibodies.

A total of 12.9% (95% CI 9.3–16.4) (n = 43) of individuals showed laboratory evidence of active infection. Anti-Treponema pallidum total antibodies (TPAb) were detected in 10 individuals (3.0%), and 33 participants (9.9%, 95% CI 6.7–13.0) tested positive for HBsAg. No cases of Human immunodeficiency virus (HIV) or hepatitis C virus (HCV) infection were identified among the evaluated participants (see Table 3).

With respect to hepatitis B (HBV), 36.4% (95% CI 31.3–41.6) (n = 122) of migrants had no prior exposure to the virus and were considered susceptible to infection. 35.8% (95% CI 30.7–41.0) (n = 120) showed evidence of past infection with acquired immunity, while 13.1% (95% CI 9.5–16.8) (n = 44) exhibited an isolated core antibody pattern, which requires further clinical assessment (see Table 4).

As part of the intervention, all participants received initial vaccine doses during the same visit, according to the accelerated vaccination protocol of the Galician Health Service (SERGAS). Specifically, 100% of participants received tetanus-diphtheria and polio vaccines, while MMR, varicella, and hepatitis B vaccines were administered to those lacking immunity based on serological results (see Table 5). Follow-up doses were scheduled through their assigned primary care centers.

## 4. Discussion

To the best of our knowledge, this study represents the first documented experience in Spain describing the outcomes of a one-stop, organized public health intervention in Galicia specifically targeting newly arrived migrants with irregular administrative status. Our findings reveal a high susceptibility to vaccine-preventable diseases within the evaluated cohort, particularly for HBV, measles, and varicella. We also identified a significant prevalence of active infections, such as HBV and syphilis, which pose important public health risks.

When compared to previous studies conducted in other Spanish regions, such as Madrid and Barcelona, our results show notable differences in seroprevalence of vaccine-preventable diseases [11,13]. For example, the proportion of individuals lacking immunity to measles was significantly higher in our cohort (22.7%) than in the studies by Norman (4.9%) and Rubio (6.9%). These discrepancies may reflect differences in sample composition, vaccination coverage in countries of origin, and length of stay in the host country. Conversely, rubella seroprevalence in our cohort was higher (95.8%) than that reported by Norman et al. (89%) and more consistent with the 94.5% found by Rubio et al. among sub-Saharan migrants [11,13]. These findings underscore the importance of interpreting migrant immunological data considering migration history, sociodemographic factors, and national and regional vaccination policies.

With respect to hepatitis B, we observed a high proportion of susceptible individuals (36.4%), which aligns with the findings of Norman et al. (32.1%). The prevalence of active HBV infection (HBsAg positive) in our sample (9.9%) was identical to that reported in the same study (9.9%) and consistent with a pooled prevalence of 10.5% among sub-Saharan refugees in a systematic review and meta-analysis [14]. These data reinforce the need to implement targeted screening and vaccination strategies for this population.

The centralized, one-stop screening approach used in this study enabled early and equitable detection of active infections, contrasting with the passive detection model described by Seedat et al., in which over 60% of cases were identified only when patients accessed the healthcare system for unrelated reasons [15]. This organizational strategy enabled structured, homogeneous care, ensuring comprehensive evaluation of all participants and reducing reliance on prior contact with the healthcare system. At this point of care, all participants underwent a basic clinical examination and medical interview as part of the intervention. Any individuals presenting symptoms or findings suggestive of active infection, dermatological conditions, untreated chronic illness (e.g., hypertension and asthma), or other medical needs were referred to the public healthcare system via their assigned primary care centers, following standard protocols in Galicia. Immediate medical attention was arranged for urgent cases, and follow-up was coordinated by the healthcare personnel responsible for the migrants’ reception facilities. This model may be replicated in other regions, provided that logistical resources, inter-institutional coordination, and appropriate healthcare staff training are ensured.

A notable finding in our cohort was the identification of an isolated core serologic profile (anti-HBc positive and HBsAg and anti-HBs negative) in 13.1% of cases, a profile not previously reported in other national studies [11,13,14]. This result may indicate past infection with loss of protective antibodies, repeated exposure, or false positives, and warrants close clinical monitoring [16]. For indeterminate serological results, vaccination was administered as a preventive measure to ensure adequate protection, considering diagnostic uncertainty and the group’s epidemiological context.

Although focused on a specific cohort, our results highlight several challenges that are likely applicable to other contexts involving the care of irregular migrant populations. This study provides a broad perspective on the epidemiological, organizational, and social implications of healthcare for migrants, aligning with European trends in migrant health [6,7,17]. The fragmentation of health information systems across Spanish regions and the lack of interoperability impedes continuity of care and contribute to missed diagnostic opportunities. These limitations—shared by other European contexts—hinder coordinated and efficient responses [18,19]. Our intervention addressed this gap by providing each migrant with a printed personal health record, facilitating follow-up of their serological and immunization status in other healthcare settings.

From a legal standpoint, the right to health for migrants is recognized in international instruments such as the International Covenant on Economic, Social and Cultural Rights [20], and in Spanish legislation, including the General Health Law and Royal Decree-Law 7/2018 [21,22]. These legal frameworks strengthen the obligation of states to ensure universal and equitable access to healthcare, including for populations with irregular administrative status.

The experience described in Galicia is aligned with recommendations from the ECDC and WHO [6,23], which advocate for culturally sensitive, participatory, and non-stigmatizing interventions to improve the acceptance and effectiveness of screening and vaccination programs in migrant populations. The adaptation of Galicia’s accelerated vaccination schedule (SERGAS) to European guidelines constitutes an effective and context-sensitive response. However, its implementation should be accompanied by ongoing evaluation to ensure impact and long-term sustainability [24]. In the medium to long term, our findings support the integration of serological screening and accelerated vaccination programs into a national public health strategy tailored to migrant population needs. Incorporating such models into healthcare planning would help improve equity in access, reduce the burden of communicable disease, and enhance the resilience of health systems in the context of increasing population mobility.

### Strengths and Limitations

One of the main strengths of this study is its successful implementation of WHO and ECDC recommendations on vaccination and screening in a real-life setting involving recently arrived undocumented migrants. By operationalizing these international guidelines through an integrated, on-site intervention, this study demonstrates the feasibility and public health value of applying global frameworks to reach vulnerable and hard-to-access populations in the early stages of migration.

It offers a comprehensive approach combining serological screening, clinical evaluation, and organizational analysis in a cohort of newly arrived migrants, providing valuable empirical evidence for public health planning. However, its cross-sectional design limits the ability to establish causal relationships, and the lack of a control group and detailed medical history data may introduce bias. Nonetheless, the findings support the need for proactive, structured, and culturally adapted interventions that guarantee the right to health and strengthen health equity for vulnerable migrant populations.

One relevant limitation of this study is the limited availability of detailed socio-demographic data. While age and sex were recorded and included in the analysis, other potentially informative variables, such as level of education, country of transit, or migration route, were inconsistently reported by participants and therefore excluded to minimize the risk of bias. The lack of such data limited our ability to explore associations between socio-demographic factors and immunological status or infection risk. Future studies should incorporate standardized tools for collecting this information in order to better characterize vulnerability profiles and tailor public health interventions accordingly.

One major limitation of this study was the exclusion of latent tuberculosis infection (LTBI) screening. Although such screening strategies have proven effective in other European contexts [25], their implementation in our setting was not feasible due to the lack of guarantees for continuity of care and treatment adherence, given the high mobility and administrative instability of the target population. The success of LTBI programs depends on multiple logistical, clinical, and social factors, which must be carefully addressed when designing public health interventions for undocumented migrants.

A crucial aspect in interpreting the results of this study is that the majority of participants originated from West African countries, particularly Mali and Senegal. This geographic origin is epidemiologically significant, as sub-Saharan Africa is a region of high endemicity for several infectious diseases, including hepatitis B virus (HBV), with an estimated prevalence of chronic infection in West Africa ranging between 9 and 10% [14,26]. European studies have reported similar HBV seroprevalence rates among African migrants [9,11,14], consistent with the 9.9% prevalence observed in our cohort. In many regions of Africa, vaccination coverage does not reach 50%, and in many cases, migrant children are either not vaccinated or only partially vaccinated, especially when they come from remote areas with poor access to immunization services [27]. These factors help explain the high proportion of individuals susceptible to vaccine-preventable diseases, such as measles, whose seroprevalence among migrants in Europe has been estimated at only 83.7%, below the herd immunity threshold [7,11]. Regarding measles, vaccination coverage in West Africa has historically been low. In Mali, for example, up to 23% of children are under-immunized, and regional coverage for the Measles-containing vaccine, 2nd dose, was only 49% in 2023, far below the 95% threshold required for herd immunity [28]. This helps explain the proportion of participants found to be serologically susceptible to measles.

This study is among the first in Spain to implement a structured, one-stop public health intervention specifically designed for newly arrived migrants in a non-urban region. By generating real-world data from Galicia, an area with limited prior evidence in this field, it contributes to filling a critical gap in seroepidemiological surveillance and demonstrates the feasibility of deploying adapted protocols beyond major metropolitan centers. Unlike studies conducted in other settings where screening and vaccination may occur separately, our intervention followed a one-stop model integrating serological assessment and immediate vaccination, in accordance with the official SERGAS accelerated vaccination protocol. This approach aligns with international recommendations for newly arrived migrants and was critical to address immunization gaps in a population with limited access to healthcare.

## 5. Conclusions and Implications

This study reveals a high susceptibility to vaccine-preventable diseases among newly arrived migrants, particularly to hepatitis B, measles, and varicella. The implementation of a one-stop, centralized intervention proved to be a feasible and effective model for delivering accelerated vaccination and systematic screening. These findings underscore the need to integrate such approaches into migrant health protocols, improve interoperability between health systems, and strengthen cultural competence among professionals to ensure equitable access to care and public health inclusion.

## Figures and Tables

**Table 1 tropicalmed-10-00169-t001:** Distribution of migrants by country of origin.

Country of Origin	n	%
Mali	182	54.2
Senegal	106	31.5
Mauritania	27	8.0
Gambia	8	2.4
Guinea	6	1.8
Côte d’Ivoire	3	0.9
Other countries	4	1.2
Total	336	100

**Table 2 tropicalmed-10-00169-t002:** Distribution of health problems identified.

Health Problems	n (%) [95% CI]
Gastrointestinal complaints	18 (20.2%) [13.2–29.7]
Skin conditions	13 (14.6%) [8.7–23.4]
Dental problems	11 (12.4%) [7.0–20.8]
Trauma-related injuries	9 (10.1%) [5.4–18.1]
Headache	8 (9.0%) [4.6–16.7]
Musculoskeletal pain	7 (7.9%) [3.9–15.4]
Respiratory symptoms	6 (6.7%) [3.1–13.9]
Ear pain (otalgia)	4 (4.5%) [1.8–11.0]
Suspected STI	2 (2.2%) [0.6–7.8]
Fever	2 (2.2%) [0.6–7.8]
Other	9 (10.1%) [5.4–18.1]
Total	89 (100%)

**Table 3 tropicalmed-10-00169-t003:** Serological results for the total sample (n = 335).

Indicator	Category	n (%) [95% CI] *
Varicella IgG	Positive	274 (82.3%) [78,9–86.0]
	Negative	55 (16.5%) [12.9–20.9]
	Indeterminate	4 (1.2%) [0.5–3.1]
Rubella IgG	Positive	321 (96.1%) [93.5–97.7]
	Negative	10 (3.0%) [1.6–5.4]
	Indeterminate	3 (0.9%) [0.3–2.6]
Measles IgG	Positive	258 (77.0%) [72.2–81.2]
	Negative	76 (22.7%) [18.5–27.5]
	Indeterminate	1 (0.3%) [0.1–1.7]
Mumps IgG	Positive	299 (90.3%) [86.7–93.1]
	Negative	30 (9.1%) [6.4–12.6]
	Indeterminate	2 (0.6%) [0.2–2.2]
TPAb	Positive	10 (3.0%) [1.6–5.4]
	Negative	325 (97.0%) [94.6–98.4]
HAV IgG	Positive	332 (99.1%) [97.4–99.7]
	Negative	3 (0.9%) [0.3–2.6]
HCV	Positive	0 (0.0%) [0.0–1.1]
	Negative	335 (100%) [98.9–100.0]
HIV	Positive	0 (0.0%) [0.0–1.1]
	Negative	335 (100%) [98.9–100.0]

* Percentages and 95% confidence intervals are calculated based on the number of individuals tested for each marker (excluding untested and indeterminate cases).

**Table 4 tropicalmed-10-00169-t004:** Serology results for the hepatitis B virus.

	HBsAg	Anti-HBc	Anti-HBs	n	%	95% CI
No infection and susceptible	-	-	-	122	36.4	31.3–41.6
Vaccinated	-	-	+	16	4.8	2.5–7.1
Past infection and immunity	-	+	+	120	35.8	30.7–41.0
Isolated core	-	+	-	44	13.1	9.5–16.8
Requires clinical assessment	+	+	-	33	9.9	6.7–13.0

**Table 5 tropicalmed-10-00169-t005:** Vaccines administered per participant *.

Vaccine	n	%
Tetanus-Diphtheria	335	100
Polio (IPV)	335	100
Meningococcal ACWY	163	48.7
MMR	137	40.9
Varicella	90	26.8
HBV	122	36.4
HAV	3	0.9

* Vaccine indication was guided by both serological results and public health recommendations [12].

## Data Availability

Data availability is under petition.

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
