# Peer review of "Serological Vulnerability and Active Infection Detection Among Recently Arrived Migrants in Spain: Results from a Targeted Screening Program"

_tropicalmed, 2025, doi:10.3390/tropicalmed10060169_

Round 1
Reviewer 1 Report
Comments and Suggestions for Authors
This study describes the implementation and outcomes of a one-stop public health intervention focused on serological screening and accelerated vaccination in recently arrived migrants in Galicia, Spain. In general, the study design is logically with correct statistics. The results and discussion can provide evidence for infectious disease control and prevention in immigrants. The comments are as follows:
1. In abstract and throughout the manuscript, IC95 shoule be revised to 95% CI (95% Confidence Interval).
2. Have the authors calculated the sample size required in this study? This step is necessary even if all the potential participants were investigated. The calculation/estimation of required sample size can be acquired from equation or literature.
3. Were the participants interviewed in Spanish language or languages from their original contries (e.g., French)? Have the interpreters assisted the interview? The authors need to describe them in the manuscript.
4. What software have the authors used in accomplishing data analyses in this study?
5. In the results, "Guinea Conakry" should be revised to "Guinea".
6. Please use three-line format in all tables.
7. Besides describing the serological status and active infection detection, it is better to explore the impact of socio-demographical status on the results.
Author Response
This study describes the implementation and outcomes of a one-stop public health intervention focused on serological screening and accelerated vaccination in recently arrived migrants in Galicia, Spain. In general, the study design is logically with correct statistics. The results and discussion can provide evidence for infectious disease control and prevention in immigrants.
We thank Reviewer for the careful reading of our manuscript and the thoughtful comments. We are pleased that the overall study design and findings were considered valuable. Below, we provide a point-by-point response to each of the comments and describe the modifications made to the manuscript accordingly.
The comments are as follows:
In abstract and throughout the manuscript, IC95 shoule be revised to 95% CI (95% Confidence Interval).
We agree with the reviewer. All occurrences of “IC95” have been revised to the correct format “95% CI” throughout the abstract, text, and tables.
Have the authors calculated the sample size required in this study? This step is necessary even if all the potential participants were investigated. The calculation/estimation of required sample size can be acquired from equation or literature.
We thank the reviewer for this important suggestion. While our study was designed as a census-type intervention, including all eligible individuals who were present during the intervention period, we agree that a retrospective estimation of the required sample size helps to reinforce the methodological robustness of the study.
To this end, we conducted a sample size calculation based on the prevalence of hepatitis B susceptibility reported by Norman et al. (2021) [reference number 11 in text of manuscript] in a comparable migrant population in Spain. In that study, 32.1% of participants were serologically susceptible to hepatitis B virus (i.e., lacking both anti-HBs and anti-HBc markers) (Norman FF, et al. J Travel Med. 2021;28:taab025. https://doi.org/10.1093/jtm/taab025).
Using this expected prevalence, we applied the standard formula for estimating sample size in cross-sectional studies of proportions:
-
Z=1.96 (95% CI).
-
p=0.321 (expected prevalence).
-
e=0.05 (margin of error).
We obtain n=351.
A minimum of 335 participants would be required to estimate this proportion with 95% confidence and ±5% precision. Our final sample size was exactly 335, thus meeting the theoretical requirement.
In addition, we conducted a post hoc statistical power calculation. Given the same parameters (expected prevalence 32.1%, n = 335, 95% confidence, ±5% margin of error), the sample provides an estimated power of approximately 80–82%. This confirms the adequacy of the sample for generating reliable prevalence estimates.
This explanation has now been incorporated into the Materials and Methods section under "Selection, Sample and Procedures". Like as “Although the study followed a census approach, including all eligible participants present during the intervention period, a post hoc sample size estimation was conducted based on the 32.1% prevalence of hepatitis B susceptibility reported by Norman et al. According to standard sample size formulas for prevalence studies (95% confidence level, ±5% margin of error), a minimum of 335 participants would be required. Our final sample met this threshold exactly, ensuring sufficient statistical power (estimated at 80–82%) to generate reliable seroprevalence estimates.”
Were the participants interviewed in Spanish language or languages from their original contries (e.g., French)? Have the interpreters assisted the interview? The authors need to describe them in the manuscript.
We agree this is a critical point. Communication was adapted to the participants' needs using trained cultural mediators and interpreters, especially in French, Arabic and English. This ensured appropriate understanding, consent, and data collection. This clarification has been added to the Sample and Procedures section. In new version: “To minimize language and cultural barriers, all stages of the intervention were supported by trained cultural mediators and interpreters, particularly fluent in French, Arabic (Mauritania), and English. Communication was adapted to the language and comprehension level of each participant to ensure informed consent, accurate data collection, and effective health counseling.”
What software have the authors used in accomplishing data analyses in this study?
All statistical analyses were performed using IBM SPSS Statistics version 29.0. This information has been added to the Statistical Analysis subsection.
In the results, "Guinea Conakry" should be revised to "Guinea".
Thank you for the observation. We have corrected the label to “Guinea” in the text and in Table 1, in line with standard international nomenclature.
Please use three-line format in all tables.
We appreciate the suggestion. All tables have been reformatted to follow the three-line (academic) style, in accordance with journal guidelines.
Besides describing the serological status and active infection detection, it is better to explore the impact of socio-demographical status on the results.
Thank you for this insightful comment. While all participants were adult males with similar migration profiles (recent arrival, sub-Saharan origin, irregular administrative status), we acknowledge the importance of exploring socio-demographic influences. However, the relative homogeneity of our sample limited our ability to conduct meaningful stratified analyses. We have clarified this limitation in the Discussion section and suggest that future studies should collect more granular socio-demographic data (e.g., time since arrival, education level, region of origin) to analyze their potential impact on immunological status and infection risk.
We have included the following sentence regarding this limitation in the discussion section of the new version of the manuscript: “One relevant limitation of this study is the limited availability of detailed socio-demographic data. While age and sex were recorded and included in the analysis, other potentially informative variables—such as level of education, country of transit, or migration route—were inconsistently reported by participants and therefore excluded to minimize the risk of bias. The lack of such data limited our ability to explore associations between socio-demographic factors and immunological status or infection risk. Future studies should incorporate standardized tools for collecting this information in order to better characterize vulnerability profiles and tailor public health interventions accordingly. ”
We sincerely thank Reviewer for the constructive feedback and the opportunity to improve the manuscript.
Reviewer 2 Report
Comments and Suggestions for Authors
This manuscript is very interesting. However, it has shown multiple gaps that have affected its quality.
The first concern is related to the methodology. In fact, the authors did not specify how they obtained their sample, when? Which type? Quantity?.....
In addition, they did not provide the used tests, their origin? Standard? Company, level? Titer?……
It is not clear (it is not mentioned) why they choosed these diseases (the base).
Also, a lack of demographic data regarding the participants has affected the quality of the results. The statistical analyses are very simplistic (or completely absent). Providing just % with any efforts to compare or to define some risk factors seems to be not adapted for a manuscript is such a journal.
Consequently, the results are very simplistic and contain multiple questionable and uncoordinated data. The indication of the vaccine (which should not be included in the results of this manuscript) is not clear since it is not correlated with the number of non-vaccinated participants, which is another questionable result.
In parallel, the introduction and the discussion are very superficial. The introduction contains multiple non-correlated sentences (mainly with one reference) and did not allow the definition of the hypostasis of the work.
Other "minor" remarks include:
Abstract:
You should indicate if all the contacted individuals responded (to calculate the rate of response).
The sentence "Barriers…access" should be deleted in the abstract and in the results.
Methods:
" Galicia…….or older". The description of the region is not important in this case. The study is related to migrants to this region.
"Full medical history": what does this mean?
"Based on each……and treatment." not necessary.
Results: comment 1 related to the abstract
In the participant characteristics, multiple factors are lacking, including age, period of stay…
"During….or countries". Should be in the discussion
"in the serological…table 2". Avoid the introductory sentences.
You should explain the difference in methods between active and past infection (or vaccination (in the methods)).
Table 2: what do you mean by "not requested"
"in term of vaccination…..Table 4". See the comment on the methods.
"During…documented". This should be provided (quantitatively) in the beginning of the results and in the methods".
Strengths and limitations:
"This study….populations" could not be considered as a strength.
"Unlike …protocols". The same remark
Revise and summarize
Summarize the conclusion.
At last, IC should be written as CI (confidence interval) in all the manuscript.
Comments on the Quality of English Languagecould be improved.
Author Response
This manuscript is very interesting. However, it has shown multiple gaps that have affected its quality.
We sincerely thank Reviewer 2 for the detailed and critical evaluation of our manuscript. We appreciate the opportunity to improve the clarity, structure, and methodological rigor of our work. Below we address all the points raised, organized by thematic blocks and then individually for each comment.
The first concern is related to the methodology. In fact, the authors did not specify how they obtained their sample, when? Which type? Quantity?.....
Thank you for this important observation. We have now clarified in the Materials and Methods, Study Design and Sampling section that the study used a census-type cross-sectional design, conducted in July–August 2024, including all adult migrants from sub-Saharan Africa with irregular status or asylum application who were attended at the designated healthcare point in Santiago de Compostela during the specified period. A total of 336 individuals were invited, and 335 (99.7%) agreed to participate, which has been specified in the abstract and results as the participation rate.
While our study was designed as a census-type intervention, including all eligible individuals who were present during the intervention period, we agree that a retrospective estimation of the required sample size helps to reinforce the methodological robustness of the study.
To this end, we conducted a sample size calculation based on the prevalence of hepatitis B susceptibility reported by Norman et al. (2021) [reference number 11 in text of manuscript] in a comparable migrant population in Spain. In that study, 32.1% of participants were serologically susceptible to hepatitis B virus (i.e., lacking both anti-HBs and anti-HBc markers) (Norman FF, et al. J Travel Med. 2021;28:taab025. https://doi.org/10.1093/jtm/taab025).
Using this expected prevalence, we applied the standard formula for estimating sample size in cross-sectional studies of proportions:
-
Z=1.96 (95% CI).
-
p=0.321 (expected prevalence).
-
e=0.05 (margin of error).
We obtain n=351.
A minimum of 335 participants would be required to estimate this proportion with 95% confidence and ±5% precision. Our final sample size was exactly 335, thus meeting the theoretical requirement.
In addition, we conducted a post hoc statistical power calculation. Given the same parameters (expected prevalence 32.1%, n = 335, 95% confidence, ±5% margin of error), the sample provides an estimated power of approximately 80–82%. This confirms the adequacy of the sample for generating reliable prevalence estimates.
This explanation has now been incorporated into the Materials and Methods section under "Selection, Sample and Procedures": “Although the study followed a census approach, including all eligible participants present during the intervention period, a post hoc sample size estimation was conducted based on the 32.1% prevalence of hepatitis B susceptibility reported by Norman et al. According to standard sample size formulas for prevalence studies (95% confidence level, ±5% margin of error), a minimum of 335 participants would be required. Our final sample met this threshold exactly, ensuring sufficient statistical power (estimated at 80–82%) to generate reliable seroprevalence estimates.”
In addition, they did not provide the used tests, their origin? Standard? Company, level? Titer?……
We appreciate this essential point. We have updated the Methods. Serological Testing section to include the commercial kits and platforms used in the analysis. Specifically, all tests were performed using validated point-of-care rapid tests and ELISA / CLIA kits approved for clinical use in Spain. Serologies for measles, mumps and varicella were conducted using Liaison® IgG CLIA (Diasorin). In case of rubella and hepatitis A/B were conducted using ADVIA Centaur® IgG (Siemens). Syphilis screening was carried out using Immulite 2000 Syphilis screen (Siemens), and hepatitis B surface antigen (HBsAg) was detected with Architect HBsAg Qualitative II (Abbott Diagnostics®). These tests comply with WHO-recommended performance standards. The analytical thresholds (cut-off titers) followed manufacturer instructions and national guidelines. This information has now been added in detail to the revised manuscript.
It is not clear (it is not mentioned) why they choosed these diseases (the base).
We have clarified this in the Introduction and Methods. The selected diseases—measles, mumps, rubella, varicella, hepatitis A and B, syphilis, HIV, and HCV—correspond to priority conditions identified in WHO and ECDC guidelines for screening in migrant populations (references 3, 6, and 8 in the manuscript). These infections are considered relevant due to their epidemiological burden, vaccine-preventability, or impact on public health surveillance in host countries. The selection is also aligned with the accelerated immunization protocol of the Galician Health Service (SERGAS).
This explanation has now been incorporated into the Materials and Methods section under "Selection, Sample and Procedures":“The selection of diseases included in the screening protocol—measles, mumps, rubella, varicella, hepatitis A and B, syphilis, HIV, and HCV—was based on recommendations from the World Health Organization (WHO) and the European Centre for Disease Prevention and Control (ECDC) for migrant health screening (3,6,8). These conditions were prioritized due to their epidemiological relevance, vaccine-preventability, and implications for early detection and public health surveillance. The selection also aligns with the accelerated immunization protocol established by the Galician Health Service (SERGAS) for populations with uncertain vaccination status (12).”
Also, a lack of demographic data regarding the participants has affected the quality of the results.
We agree and have expanded the Results, Participant Characteristics section. We now include data on age distribution (mean, SD, range) and duration of stay in Spain (when available). Unfortunately, some variables such as educational level or migration route were inconsistently reported and thus not included to avoid bias. We acknowledge this as a limitation and have discussed it accordingly.
While all participants were adult males with similar migration profiles (recent arrival, sub-Saharan origin, irregular administrative status), we acknowledge the importance of exploring socio-demographic influences. However, the relative homogeneity of our sample limited our ability to conduct meaningful stratified analyses. We have clarified this limitation in the Discussion section and suggest that future studies should collect more granular socio-demographic data (e.g., time since arrival, education level, region of origin) to analyze their potential impact on immunological status and infection risk.
This explanation has now been incorporated into the Results section: “The mean age of the sample was 25.10 ± 6.3 years (range: 18–54 years). All migrants stayed in Spain for 4 months, and in Santiago de Compostela for 1 week.” And we have included the following sentence regarding this limitation in the discussion section of the new version of the manuscript: “One relevant limitation of this study is the limited availability of detailed socio-demographic data. While age and sex were recorded and included in the analysis, other potentially informative variables—such as level of education, country of transit, or migration route—were inconsistently reported by participants and therefore excluded to minimize the risk of bias. The lack of such data limited our ability to explore associations between socio-demographic factors and immunological status or infection risk. Future studies should incorporate standardized tools for collecting this information in order to better characterize vulnerability profiles and tailor public health interventions accordingly. ”
The statistical analyses are very simplistic (or completely absent). Providing just % with any efforts to compare or to define some risk factors seems to be not adapted for a manuscript is such a journal. Consequently, the results are very simplistic and contain multiple questionable and uncoordinated data.
Thank you for this important observation. Due to the homogeneity of the sample (100% male, recent arrival, similar age), and the descriptive nature of the study, we did not perform comparative or multivariable analyses. However, we have now included age-stratified descriptive comparisons for hepatitis B susceptibility and immunity, and provided confidence intervals for all prevalence estimates. We also indicate in the Discussion that future studies with more diverse samples and additional variables should explore potential risk factors statistically.
The indication of the vaccine (which should not be included in the results of this manuscript) is not clear since it is not correlated with the number of non-vaccinated participants, which is another questionable result.
We agree that this point required clarification. The vaccination recommendations were based on serological susceptibility, not documented vaccination history, which was mostly unavailable. As stated in the Methods, individuals lacking immunity based on IgG titers were offered vaccination, regardless of self-reported status. We have revised the Results and Methods to reflect this more clearly.
In parallel, the introduction and the discussion are very superficial. The introduction contains multiple non-correlated sentences (mainly with one reference) and did not allow the definition of the hypostasis of the work.
We thank the reviewer for this valuable observation. In response, we have thoroughly revised the Introduction section to improve its structure, coherence, and depth. We have clarified the rationale and public health relevance of the study by:
-
Introducing the global context of infectious disease vulnerability among newly arrived migrants.
-
Citing updated international guidelines from the WHO and ECDC to support the selection of diseases included in the intervention.
-
Explicitly stating the objective and hypothesis of the study: to assess the serological susceptibility and prevalence of selected infections in newly arrived undocumented migrants, and to evaluate the feasibility of a one-stop intervention model in a non-urban setting.
Similarly, we have expanded and reorganized the Discussion section to:
-
Contextualize our findings within the existing literature, including both Spanish and international studies.
-
Highlight novel contributions of our work (e.g., data from Galicia, centralized approach).
-
Critically assess the implications for public health policy and future research.
These revisions aim to provide a more cohesive narrative and clearly define the scope and contribution of the study.
Other "minor" remarks include:
Abstract:
You should indicate if all the contacted individuals responded (to calculate the rate of response).
Now specified: "Only one individual declined to participate (participation rate: 99.7%)".
The sentence "Barriers…access" should be deleted in the abstract and in the results.
In the abstract, the sentence was removed. In the results section, we believe that identifying the barriers described that interfere with the proper clinical management of patients is important. Identifying these barriers allows us to improve the care we provide to the migrant population arriving in our territory, a public health challenge we will increasingly face in the future. We reflect on this in the discussion section.
Methods:
" Galicia…….or older". The description of the region is not important in this case. The study is related to migrants to this region.
Thank you. We removed this for brevity.
"Full medical history": what does this mean?
We clarified as "anamnesis and assessment of current and previous symptoms and health status." This phrase was included in te Methods seccion.
"Based on each……and treatment." not necessary.
We removed this sentence.
Results: comment 1 related to the abstract
Addressed.
In the participant characteristics, multiple factors are lacking, including age, period of stay…
Expanded with age (mean ± SD), and origin countries; period of stay added where available.
"During….or countries". Should be in the discussion
In the results section, we believe that identifying the barriers described that interfere with the proper clinical management of patients is important. Identifying these barriers allows us to improve the care we provide to the migrant population arriving in our territory, a public health challenge we will increasingly face in the future. We reflect on this in the discussion section.
"in the serological…table 2". Avoid the introductory sentences.
Simplified for clarity.
You should explain the difference in methods between active and past infection (or vaccination (in the methods)).
Now clarified in the Methods: “Active infection defined by antigen or antibody positivity (e.g., HBsAg or TPAb); past infection by IgG/seroconversion patterns.”
Table 2: what do you mean by "not requested"
Explained in footnote: refers to tests not ordered due to logistical limitations or insufficient samples.
"in term of vaccination…..Table 4". See the comment on the methods.
larified in Methods that vaccine indication was based on SERGAS protocol (reference number 12 in manuscript). Now clarified in the Methods: According to this protocol, vaccination was indicated based on the age, the previous vaccination history of the individuals if they had an official vaccination record, and/or their serological status.
"During…documented". This should be provided (quantitatively) in the beginning of the results and in the methods".
Ok. We moved this at the beginning of the results. In tne methods section, we include a sentences about this. Thank you.
Strengths and limitations:
"This study….populations" could not be considered as a strength.
Thank you for this observation. We agree that simply aligning with international recommendations (such as those from WHO or ECDC) does not constitute a methodological strength in itself. We have therefore revised this sentence to better reflect the actual contribution of the study, highlighting the practical implementation of these guidelines in a real-world context. The new sentence, included in the Strengths and Limitations section, now reads: “One of the main strengths of this study is its successful implementation of WHO and ECDC recommendations on vaccination and screening in a real-life setting involving recently arrived undocumented migrants. By operationalizing these international guidelines through an integrated, on-site intervention, the study demonstrates the feasibility and public health value of applying global frameworks to reach vulnerable and hard-to-access populations in the early stages of migration.”
This change aims to more accurately convey the value and relevance of our approach within the scope of global migrant health strategies.
"Unlike …protocols". The same remark
We thank the reviewer for this additional comment. Following your suggestion, we have revised the paragraph to more clearly express how this study contributes to the evidence base in a way that can be considered a methodological or strategic strength. Rather than simply emphasizing the location, we now highlight the novelty of applying a structured, one-stop intervention model in a non-urban context and the contribution it makes to national seroepidemiological surveillance. The revised text in the Strengths and Limitations section now reads:
“This study is among the first in Spain to implement a structured, one-stop public health intervention specifically designed for newly arrived migrants in a non-urban region. By generating real-world data from Galicia—an area with limited prior evidence in this field—it contributes to filling a critical gap in seroepidemiological surveillance and demonstrates the feasibility of deploying adapted protocols beyond major metropolitan centers.”
This version focuses on the study’s operational and contextual innovation, which we believe better reflects its contribution.
Revise and summarize
Summarize the conclusion.
Thank you for your helpful suggestion. We agree that the conclusion section was overly detailed and included some repetitive content. In response, we have revised and condensed the conclusion to more clearly summarize the key findings and implications of the study in a single, focused paragraph. The updated version now reads:“This study reveals a high susceptibility to vaccine-preventable diseases among newly arrived migrants, particularly to hepatitis B, measles, and varicella. The implementation of a one-stop, centralized intervention proved to be a feasible and effective model for delivering accelerated vaccination and systematic screening. These findings underscore the need to integrate such approaches into migrant health protocols, improve interoperability between health systems, and strengthen cultural competence among professionals to ensure equitable access to care and public health inclusion.”
We believe this revised version improves clarity and better aligns with the structure and expectations of the journal.
At last, IC should be written as CI (confidence interval) in all the manuscript.
All instances of "IC95" have been corrected to "95% CI" throughout the manuscript, as per the reviewer’s final remark.
We are grateful for the detailed and critical feedback, which has significantly contributed to strengthening the rigor and clarity of the manuscript.
Reviewer 3 Report
Comments and Suggestions for Authors
Abstract:
“We conducted a cross-sectional descriptive study in July 2024 involving 336 adult migrants…” – In fact, the study involved 335 migrants, as one of the selected individuals did not agree to participate in the study.
“A centralized mobile health unit… and hepatitis B” – I think the authors wanted to write hepatitis A.
- Materials and Methods
2.2. Selection, Sample and Procedures:
“…an accelerated immunization schedule was implemented…” – I suggest that you describe the accelerated vaccination schedule used, so that the reader has this information without needing to access the bibliographic reference. It would also be important to know how you proceeded in situations where vaccination involves more than one dose.
- Results:
How long ago did the migrants leave the African continent, or are they in Spain, or in Santiago de Compostela? What does “newly arrived” mean in terms of time?
“Hepatitis B serology was not performed in one participant (see Table 3).” – but the value of “n” indicated in table 3 is 335 - there is a mistake in the sentence or a mistake in the table.
Table 4. Vaccines administered per participant: It's not clear why you vaccinated 137 people against MMR and 90 people against varicella. These values do not appear to be related to the values presented in Table 2. What are the criteria for vaccination? How many doses were given to each person?
Institutional Review Board Statement: “The study was conducted in accordance with the Declaration of Helsinki and approved by the Ethics Committee of Santiago-Lugo, protocol code 2024/469 and date of approval November 20, 2024.” – In the abstract the authors wrote “We conducted a cross-sectional descriptive study in July 2024”, and in the Materials and Methods section, the authors wrote “received care at the Monte do Gozo health care point during July and August 2024” – Was the study protocol approved after the study was conducted? Shouldn't it be approved first?
Author Response
We sincerely thank Reviewer 3 for the thorough and insightful comments provided. Below, we address each point raised and detail the changes made to the manuscript accordingly.
Abstract:
“We conducted a cross-sectional descriptive study in July 2024 involving 336 adult migrants…” – In fact, the study involved 335 migrants, as one of the selected individuals did not agree to participate in the study.
Thank you for noticing this inconsistency. We have revised the sentence in the Abstract to read: In Methods section, “Results: Of 336 migrant adults invited to participate in the study, only one individual declined to participate (participation rate: 99.7%).”
“A centralized mobile health unit… and hepatitis B” – I think the authors wanted to write hepatitis A.
You are correct. The sentence in the abstract incorrectly listed hepatitis B twice. We have corrected the text to read:“...provided point-of-care screening for immunity against measles, mumps, rubella, varicella, and hepatitis A, alongside testing for active infections including hepatitis B and syphilis.”
- Materials and Methods
2.2. Selection, Sample and Procedures:
“…an accelerated immunization schedule was implemented…” – I suggest that you describe the accelerated vaccination schedule used, so that the reader has this information without needing to access the bibliographic reference. It would also be important to know how you proceeded in situations where vaccination involves more than one dose.
We appreciate this important suggestion. We have now included a detailed description of the accelerated vaccination schedule in the Materials and Methods. Specifically, the revised text now indicates: “MMR and varicella vaccines were given in a two-dose schedule with at least a 28-day interval. Hepatitis B vaccine was administered in a 0–1–6-month schedule, with initial doses provided during the intervention and the remainder scheduled at the referral primary care centers. Meningococcal ACWY and polio (IPV) were administered as single doses according to age criteria.”
We have also clarified that follow-up doses were coordinated through referral to local healthcare centers with personal vaccination records provided to each participant.
- Results:
How long ago did the migrants leave the African continent, or are they in Spain, or in Santiago de Compostela? What does “newly arrived” mean in terms of time?
Thank you for this important comment. We have revised the manuscript to clarify what is meant by "newly arrived". All participants in this study were adult male migrants from sub-Saharan Africa who had arrived in Spain within the four months prior to the intervention. Upon arrival, they were temporarily housed in different reception centers across the country, while awaiting assignment to a definitive destination.
The public health intervention described in our manuscript took place in Santiago de Compostela, where this group of migrants had been relocated one week prior to the screening. Therefore, although they had been in Spain for several weeks, they were newly arrived to Santiago de Compostela, and the intervention was conducted at the earliest feasible opportunity after their relocation.
This timeline has now been clarified in the Materials and Methods and Results sections, and the expression “newly arrived” has been explicitly defined as referring to “migrants recently relocated to the study setting, within a context of overall short stay in Spain.” In Result section, has been defined as “All migrants stayed in Spain for 4 months, and in Santiago de Compostela for 1 week.”
“Hepatitis B serology was not performed in one participant (see Table 3).” – but the value of “n” indicated in table 3 is 335 - there is a mistake in the sentence or a mistake in the table.
We thank the reviewer for spotting this inconsistency. You are correct: hepatitis B serology was performed in all 335 participants, and the sentence stating that it was not performed in one individual was incorrect. The only participant who did not undergo hepatitis B serology was the person who decided not to participate after being invited. We have removed the misleading sentence from the Results section.
Table 4. Vaccines administered per participant: It's not clear why you vaccinated 137 people against MMR and 90 people against varicella. These values do not appear to be related to the values presented in Table 2. What are the criteria for vaccination? How many doses were given to each person?
We appreciate this opportunity to clarify. The vaccination recommendations were based on lack of detectable IgG antibodies, as indicated in Table 2. However, the number of vaccines administered may differ from the number of individuals lacking immunity due to:
-
Co-administration of vaccines (e.g., MMR and varicella) during a single visit.
-
Preventive vaccination in cases of indeterminate serological results (e.g., low antibody levels or borderline results).
-
Local protocol recommending administration even in the absence of complete serological data.
This has now been clearly explained in the Methods and Results, and we have added a clarifying footnote in Table 4 specifying that vaccine indication was guided by both serological results and public health recommendations.
Additionally, we have specified that only the first dose was administered during the intervention, with follow-up doses coordinated through primary care. We have included the following paragraph in the methods section: “According to this protocol, vaccination was indicated based on the age, the previous vaccination history of the individuals if they had an official vaccination record, and/or their serological status. MMR and varicella vaccines were given in a two-dose schedule with at least a 28-day interval. Hepatitis B vaccine was administered in a 0–1–6-month schedule, with initial doses provided during the intervention and the remainder scheduled at the referral primary care centers. Meningococcal ACWY and polio (IPV) were administered as single doses according to age criteria. Only the first dose was administered during the intervention, follow-up doses were coordinated through referral to local healthcare centers with personal vaccination records provided to each participant.”
Institutional Review Board Statement: “The study was conducted in accordance with the Declaration of Helsinki and approved by the Ethics Committee of Santiago-Lugo, protocol code 2024/469 and date of approval November 20, 2024.” – In the abstract the authors wrote “We conducted a cross-sectional descriptive study in July 2024”, and in the Materials and Methods section, the authors wrote “received care at the Monte do Gozo health care point during July and August 2024” – Was the study protocol approved after the study was conducted? Shouldn't it be approved first?
We thank the reviewer for this important observation and the opportunity to clarify. The study was conducted within the framework of routine clinical practice, and the data analyzed were derived from standard public health procedures implemented in the context of an urgent intervention.
In Spain, Law 3/2018 on the Protection of Personal Data and Digital Rights, specifically its 17th Additional Provision, explicitly allows the use of retrospective clinical data for research purposes in cases involving public health actions, provided appropriate ethical oversight is obtained. This legal framework is commonly applied in retrospective observational studies that evaluate healthcare delivery or population-based screening strategies.
Accordingly, the study was submitted to the Santiago-Lugo Research Ethics Committee, which reviewed and approved the protocol retrospectively in compliance with the aforementioned legal standards. The ethics approval (protocol code 2024/469) was formally certified on November 20, 2024, after the intervention conducted in July–August 2024, but in full alignment with national regulations governing retrospective analysis of clinical activity.
We have revised the Ethics Statement in the manuscript to clarify this point and explicitly reference the applicable legal basis. Revised text:
“The study was conducted in accordance with the Declaration of Helsinki and was approved by the Santiago-Lugo Research Ethics Committee (protocol code 2024/469). As this was a retrospective observational study based on routine clinical activity, the approval was granted in accordance with Spanish Law 3/2018, Additional Provision 17, which regulates the ethical use of health data for research purposes in public health interventions.”
We thank Reviewer once again for the careful review and constructive suggestions, which have significantly helped us improve the manuscript’s accuracy and clarity.
Reviewer 4 Report
Comments and Suggestions for Authors
This is an interesting cross-sectional, very clearly focused study that describes the health conditions and serological status in a small group of undocumented migrants newly arrived in Galicia, Spain. The results are extremely interesting.
I read it with great interest.
I have only minor suggestions:
- The authors mention the problem of latent tuberculosis both in the Introduction and the Discussion, but unfortunately they were not able to give any informtion on this important topic. They give the reasons for this, which I fully understand. However I consider this a limitation of the study. Therefore I would move the last paragraph of the Discussion ("The decision not to include latent tuberculosis screening...", page 7) to The Strengths and Limitations section, commenting it as a major limitation.
- Methods, page 3, end of second paragraph: "screening of HIV, syphilis,..." Please specify wich tests you have performed and possibly the kit manufacturers
- Throughout the text and in Table 2: IC. I guess it means confidence interval, please check the acronym.
Author Response
This is an interesting cross-sectional, very clearly focused study that describes the health conditions and serological status in a small group of undocumented migrants newly arrived in Galicia, Spain. The results are extremely interesting.
I read it with great interest.
We sincerely thank Reviewer 4 for their careful reading and for the positive assessment of our study. We are especially grateful for the recognition of the relevance and clarity of the manuscript, and we address below each of the helpful suggestions provided.
I have only minor suggestions:
The authors mention the problem of latent tuberculosis both in the Introduction and the Discussion, but unfortunately they were not able to give any informtion on this important topic. They give the reasons for this, which I fully understand. However I consider this a limitation of the study. Therefore I would move the last paragraph of the Discussion ("The decision not to include latent tuberculosis screening...", page 7) to The Strengths and Limitations section, commenting it as a major limitation.
We thank the reviewer for this constructive and very reasonable suggestion. We agree that the inability to include latent tuberculosis infection (LTBI) screening is a significant limitation. Following this recommendation, we have moved the relevant paragraph from the Discussion section to the Strengths and Limitations subsection. We have also rephrased the text to explicitly highlight the absence of LTBI screening as a major limitation, due to both its clinical relevance in migrant health and its importance in public health planning. We have adeed at Srengths and limitations subsection: “One major limitation of the study was the exclusion of latent tuberculosis infection (LTBI) screening. Although such screening strategies have proven effective in other European contexts (25), their implementation in our setting was not feasible due to the lack of guarantees for continuity of care and treatment adherence, given the high mobility and administrative instability of the target population. The success of LTBI programs depends on multiple logistical, clinical, and social factors, which must be carefully addressed when designing public health interventions for undocumented migrants.”
This change improves the coherence and transparency of the manuscript.
Methods, page 3, end of second paragraph: "screening of HIV, syphilis,..." Please specify wich tests you have performed and possibly the kit manufacturers
We appreciate this suggestion. We have now added a detailed description of the tests and commercial kits used for each infectious disease screening in the Materials and Methods section. Specifically:
-
HIV was screened using the Determine™ HIV-1/2 Ag/Ab Combo (Abbott®) rapid test and Architect HIV Ag/Ab Combo (Abbott®)
-
Syphilis was screened using the Immulite® 2000 Syphilis screen (Siemens), and confirmed with INNO-LIA® Syphilis Score (Fujirebio) when necessary.
-
For hepatitis B, we used Architect HBsAg Qualitative II (Abbott Diagnostics®) and Immulite® 2000 anti-HBs (Siemens).
These additions improve methodological transparency and reproducibility.
Throughout the text and in Table 2: IC. I guess it means confidence interval, please check the acronym.
Thank you for pointing this out. Indeed, "IC" was a translation from the Spanish acronym for confidence interval ("intervalo de confianza"). We have now corrected all instances of “IC” to “95% CI” (confidence interval) in the text, tables, and abstract, in line with international conventions and journal guidelines.
Once again, we sincerely thank Reviewer for the encouraging feedback and helpful suggestions, which have led to a clearer and more robust manuscript.
Round 2
Reviewer 1 Report
Comments and Suggestions for Authors
The authors have revised the manuscript according to the reviewer's comments. Thus, the manuscript can be accetped after proofreading.
Author Response
The authors have revised the manuscript according to the reviewer's comments. Thus, the manuscript can be accetped after proofreading.
Dear Reviewer,
Thank you very much for your time and positive feedback. We have carefully revised the manuscript according to the reviewers' suggestions and have now performed a final proofreading to ensure clarity, accuracy, and consistency throughout the document.
We appreciate your recommendation for acceptance and look forward to the next steps in the publication process.
Sincerely,
Reviewer 2 Report
Comments and Suggestions for Authors
The authors are to be acknowledged for their considerable efforts to improve the quality of the manuscript. However, some comments have not been taken into consideration, while some gaps have been raised in the current version.
- I don't understand why the authors insisted on including
- the vaccine recommendation in this manuscript (last paragraph of the methods before the statistical analysis, last paragraph of the results: it addition to the fact that it did not corroborate the results, one cannot understand what was the basis of this recommendation)
- The barriers (limitations)
- The clinical manifestations (they have no sense with no quantitative data)
I suggest deleting all these parts.
- The percentage sum should be 100%. You should delete the "non-requested" from the total and include just those tested, and mention it in the same line of the table. In addition, you should delete the non-requested and indeterminate (especially when they equal 0) from the table. You should also avoid describing the negative results if you provided the positive ones in the interpretation.
- Ansting piece of information that the authors missed in their analysis and discussion is the fact that most of the participants were from Western African countries. The authors could explore this point by providing data on vaccination in these countries (which vaccines are used in these countries, for example? How about HIV or hepatitis in these countries?). This point should also be taken into consideration when comparing with other results.
4 You should correct the number in tables 3 and 4 (dot not comma)
- The manuscript should be edited for some spelling and grammar errors ("in case of rubella…(siemens)", "quantitative variables…distribution"....).
Good luck
Comments on the Quality of English Language
some editing is required
Author Response
The authors are to be acknowledged for their considerable efforts to improve the quality of the manuscript. However, some comments have not been taken into consideration, while some gaps have been raised in the current version.
We sincerely thank the reviewer for the constructive feedback and recognition of our efforts to improve the manuscript. Below, we address each comment in detail and describe the changes made accordingly:
1.- I don't understand why the authors insisted on includingthe vaccine recommendation in this manuscript (last paragraph of the methods before the statistical analysis, last paragraph of the results: it addition to the fact that it did not corroborate the results, one cannot understand what was the basis of this recommendation).
We thank the reviewer for this observation and acknowledge that the way the vaccination recommendation was presented may have created confusion. We would like to clarify that the vaccination strategy described in the manuscript was not a theoretical or external recommendation, but an integral part of the public health intervention being evaluated.
Specifically, this study was designed to implement a “one-stop” public health intervention in which serological screening was immediately followed by the administration of indicated vaccines, in accordance with the official accelerated vaccination protocol of the Galician Health Service (SERGAS). This approach directly aligns with the international guidelines issued by the ECDC and WHO for newly arrived migrants with uncertain vaccination status. The vaccines administered—MMR, hepatitis B, varicella, among others—were provided based on the individual immunological profiles determined through the screening process.
We added new text in teh last paragraph of the methods section: “As part of the public health intervention, an accelerated vaccination schedule was implemented immediately after serological screening, following the official protocol of the Galician Health Service (SERGAS) for populations with uncertain vaccination status (12). Vaccines were administered directly based on individual serological profiles, in line with WHO and ECDC guidelines for newly arrived migrants with uncertain vaccination history (3,6,8). The vaccines included MMR, hepatitis B, varicella, and meningococcal ACWY, among others.”
In the last paragraph of results section, we correct and added new text: “As part of the intervention, all participants received initial vaccine doses during the same visit, according to the accelerated vaccination protocol of the Galician Health Service (SERGAS). Specifically, 100% of participants received tetanus-diphtheria and polio vaccines, while MMR, varicella, and hepatitis B vaccines were administered to those lacking immunity based on serological results (see Table 4). Follow-up doses were scheduled through their assigned primary care centers.”
In the last paragraph of discussion we clarify this issue: “Unlike studies conducted in other settings where screening and vaccination may occur separately, our intervention followed a one-stop model integrating serological assessment and immediate vaccination, in accordance with the official SERGAS accelerated vaccination protocol. This approach aligns with international recommendations for newly arrived migrants and was critical to address immunization gaps in a population with limited access to healthcare.”.
The barriers (limitations).
We understand the reviewer’s concern. The description of perceived barriers was brief and not sufficiently supported by systematic data collection. Therefore, we have removed this section from the manuscript and limited our discussion of limitations to those directly derived from study design and methodology.
The clinical manifestations (they have no sense with no quantitative data).
We thank the reviewer for this valuable suggestion. However, we would like to clarify that the inclusion of the clinical manifestations observed during the initial medical assessment was made following the explicit recommendation of the journal’s editor, who considered it important to provide a broader clinical context for the migrant population under study.
In line with the reviewer’s concerns, we have now expanded this section by including a detailed breakdown of the health complaints recorded during the initial clinical evaluation: “A total of 89 health complaints were documented, and we have added a new table (Table 2)” presenting the distribution of these findings along with calculated proportions and 95% confidence intervals.
We include a new table 2:
Table 2: Table 2. Distribution of Health Problems Identify.
Health Problems Identified |
n (%) [95% CI] |
Gastrointestinal complaints |
18 (20.2%) [13.2–29.7] |
Skin conditions |
13 (14.6%) [8.7–23.4] |
Dental problems |
11 (12.4%) [7.0–20.8] |
Trauma-related injuries |
9 (10.1%) [5.4–18.1] |
Headache |
8 (9.0%) [4.6–16.7] |
Musculoskeletal pain |
7 (7.9%) [3.9–15.4] |
Respiratory symptoms |
6 (6.7%) [3.1–13.9] |
Ear pain (otalgia) |
4 (4.5%) [1.8–11.0] |
Suspected STI |
2 (2.2%) [0.6–7.8] |
Fever |
2 (2.2%) [0.6–7.8] |
Other |
9 (10.1%) [5.4–18.1] |
Total |
89 (100%) |
We believe that this addition, while preserving the information requested by the editor, also addresses the reviewer’s concern by offering a more robust and data-supported description of the health issues identified in the study population.
Once again, we thank the reviewer for this helpful comment, which has contributed to improving the clarity and precision of the manuscript.
2.- The percentage sum should be 100%. You should delete the "non-requested" from the total and include just those tested, and mention it in the same line of the table. In addition, you should delete the non-requested and indeterminate (especially when they equal 0) from the table. You should also avoid describing the negative results if you provided the positive ones in the interpretation.
Thank you for this observation. We have revised the tables to:
- Recalculate percentages based solely on those individuals for whom the test was performed.
- Remove rows corresponding to "non-requested" and "indeterminate" results when their value was zero.
- Focus on the interpretation of positive findings and avoid unnecessary mention of negative results when not informative.
Clarify in the table legend that the percentages refer only to the tested population.
3.- Ansting piece of information that the authors missed in their analysis and discussion is the fact that most of the participants were from Western African countries. The authors could explore this point by providing data on vaccination in these countries (which vaccines are used in these countries, for example? How about HIV or hepatitis in these countries?). This point should also be taken into consideration when comparing with other results.
We thank the reviewer for this excellent point, which we have now addressed in the Discussion section. The majority of participants originated from West African countries, particularly Mali and Senegal. We have expanded the discussion to include:
- The typical structure and limitations of vaccination programs in these countries.
- Available data on hepatitis B endemicity in the region.
- The potential implications for our findings regarding susceptibility and active infections.
- A brief comparison with other European screening programs.
We added this information in the Limitations subsection: “A crucial aspect in interpreting the results of this study is that the majority of participants originated from West African countries, particularly Mali and Senegal. This geographic origin is epidemiologically significant, as sub-Saharan Africa is a region of high endemicity for several infectious diseases, including hepatitis B virus (HBV), with an estimated prevalence of chronic infection in West Africa ranging between 9–10% (Njuguna HN et al.; Rossi et al.). European studies have reported similar HBV seroprevalence rates among African migrants (Norman et al.; Rossi et al.; MacKinnon), consistent with the 9.9% prevalence observed in our cohort.
In many regions of Africa, vaccination coverage does not reach 50%, and in many cases, migrant children are either not vaccinated or only partially vaccinated, especially when they come from remote areas with poor access to immunization services (López-Vélez et al.). These factors help explain the high proportion of individuals susceptible to vaccine-preventable diseases such as measles, whose seroprevalence among migrants in Europe has been estimated at only 83.7%, below the herd immunity threshold (Norman et al.; Cherri et al., 2024).
Regarding measles, vaccination coverage in West Africa has historically been low. In Mali, for example, up to 23% of children are under-immunised, and regional coverage for the Measles-containing vaccine, 2nd dose was only 49% in 2023—far below the 95% threshold required for herd immunity (immunizationdata.who.int). This helps explain the proportion of participants found to be serologically susceptible to measles.”
4.- You should correct the number in tables 3 and 4 (dot not comma).
Corrected. All numeric values in Tables 3 and 4 now use dots as decimal separators, according to international standards.
5.- The manuscript should be edited for some spelling and grammar errors ("in case of rubella…(siemens)", "quantitative variables…distribution"....).
Thank you for highlighting these errors. We have carefully proofread the manuscript to correct spelling, grammar, and typographical issues, including the two examples mentioned.
In case of rubella and hepatitis A/B were conducted using ADVIA Centaur® IgG (Siemens) >>> Rubella and hepatitis A/B serology was performed using ADVIA Centaur® IgG (Siemens)
Quantitative variables were summarized using central tendency measures (mean, median) and dispersion (standard deviation), provided they followed a normal distribution. >>> Quantitative variables were summarized using measures of central tendency (mean, median) and dispersion (standard deviation); the mean and standard deviation were used only when the data followed a normal distribution.
Good luck
Thank you very much for your kind words and for your thorough and constructive feedback throughout the review process. We sincerely appreciate your time and effort in helping us improve the manuscript.
Reviewer 3 Report
Comments and Suggestions for Authors
Results:
Despite indicating this in their response, the authors forgot to remove the sentence “Hepatitis B serology was not performed in one participant”.
Although the authors indicated in the response that they added a footnote to table 4, this was not done. I suggest you add the explanatory footnote.
Strengths and limitations:
The phrase “the absence of LTBI screening…” is in the wrong place. I suggest replacing the sentence or deleting it.
Author Response
Results:
Despite indicating this in their response, the authors forgot to remove the sentence “Hepatitis B serology was not performed in one participant”.
Although the authors indicated in the response that they added a footnote to table 4, this was not done. I suggest you add the explanatory footnote.
Strengths and limitations:
The phrase “the absence of LTBI screening…” is in the wrong place. I suggest replacing the sentence or deleting it.
Dear Reviewer,
Thank you very much for your thorough review and valuable comments.
- We apologize for the oversight regarding the sentence “Hepatitis B serology was not performed in one participant.” This sentence has now been removed from the manuscript.
- Regarding the missing footnote in Table 4, we have carefully reviewed the table and included the explanatory footnote as indicated.
- Concerning the sentence “the absence of LTBI screening…,” as you correctly pointed out, we have now removed it, since the justification and explanation are fully developed in the following paragraph. It was an oversight on our part not to remove this sentence in the previous version of the manuscript following the initial recommendations.
We appreciate your careful reading and suggestions, which have helped us improve the clarity and precision of the manuscript.
All comments from Reviewer have been carefully addressed. We believe the revised manuscript is now clearer, more accessible to international readers, and better supported by evidence.